# Sequence Length of HIV-1 Subtype B Increases over Time: Analysis of a Cohort of Patients with Hemophilia over 30 Years

**DOI:** 10.3390/v13050806

**Published:** 2021-04-30

**Authors:** Young-Keol Cho, Jung-Eun Kim, Brian T. Foley

**Affiliations:** 1Department of Microbiology, Asan Medical Center, University of Ulsan College of Medicine, Seoul 05505, Korea; kimje2000@nate.com; 2HIV Databases, Theoretical Biology and Biophysics Group, Los Alamos National Laboratory, Los Alamos, NM 87544, USA; btf@lanl.gov

**Keywords:** full-length coding region sequence, HIV-1, Korean subclade B, sequence length, hemophilia, evolution

## Abstract

We aimed to investigate whether the sequence length of HIV-1 increases over time. We performed a longitudinal analysis of full-length coding region sequences (FLs) during an HIV-1 outbreak among patients with hemophilia and local controls infected with the Korean subclade B of HIV-1 (KSB). Genes were amplified by overlapping RT-PCR or nested PCR and subjected to direct sequencing. Overall, 141 FLs were sequentially determined over 30 years in 62 KSB-infected patients. Phylogenetic analysis indicated that within KSB, two FLs from plasma donors O and P comprised two clusters, together with 8 and 12 patients with hemophilia, respectively. Signature pattern analysis of the KSB of HIV-1 revealed 91 signature nucleotide residues (1.1%). In total, 48 and 43 signature nucleotides originated from clusters O and P, respectively. Six positions contained 100% specific nucleotide(s) in clusters O and *P*. In-depth FL analysis for over 30 years indicated that the KSB FL significantly increased over time before combination antiretroviral therapy (cART) and decreased with cART. This increase occurred due to the significant increase in *env* and *nef* genes, originating in the variable regions of both genes. The increase in sequence length of HIV-1 over time suggests an evolutionary direction.

## 1. Introduction

We previously conducted a nationwide genetic analysis of HIV-1 using sera from individuals in the early stages of HIV-1 infection (before 1994) to identify the cause of the HIV-1 outbreak among patients with hemophilia in Korea in 1990–1994. These molecular epidemiological studies revealed that the viruses from two cash-paid plasma donors were incompletely inactivated in the process of manufacturing clotting factor IX and were identified as the agents of infection among 20 HIV-1-infected patients with hemophilia [1,2,3,4,5,6]. The viruses in 8 and 12 patients with hemophilia infected with the Korean subclade of HIV-1 subtype B (KSB) originated from plasma donors O and P, respectively. In these studies [1,2,3,4,5,6,7], we conducted an in-depth genetic analysis, but a few genes are yet to be explored. The sequence length significantly affects the extent of clustering in a phylogenetic tree [8,9]. KSB is a distinct, monophyletic clade within the HIV-1 subtype B and is presumed to have originated from strains in the USA through a founder effect [8,10,11,12,13]. The most recent common ancestor virus is estimated to have been active around 1984 [14]; however, the earliest infection case was diagnosed in 1988 [2,3,4,5,6].

In this study, we identified signature pattern residues in the full-length coding region sequence (FL) of KSB. We also performed a phylogenetic analysis at the FL level in 64 patients, including 20 patients with hemophilia B (HPs) [1,2,3,4]. We confirmed a previously postulated epidemiological link between the viruses that infected 20 HPs and 2 plasma donors and the viruses that infected local control patients [2,3,4,5,6]. Additionally, their longitudinal sequence analyses conducted over approximately 30 years indicate that the sequence length in FL significantly increases over time before the administration of combination antiretroviral therapy (cART). It has been reported that the progression in subtype D-infection is four-fold faster than that in subtype-B infections [15]. The sequence length of subtype D is the shortest among group M.

To date, our study is one of the most comprehensive and longest longitudinal studies conducted to examine the evolution of the HIV-1 subtype B originated from a single source of HIV-1 [1,2,3,4,5,6], and its results provide novel insights into the pathogenesis of HIV-1 infection over time.

## 2. Materials and Methods

### 2.1. Ethical Statement

This study was approved by the institutional review board of the Asan Medical Center (Code 2012-0390, 4 June 2012). All subjects provided their informed consent for inclusion before participating in the study. The study was conducted in accordance with the Declaration of Helsinki.

### 2.2. Patients and Samples

Four HIV-1 infected plasma donors were diagnosed during primary infection in 1990–1992. Their plasma was used to manufacture the domestic clotting factor IX. Viruses from donors O and *P* were incompletely inactivated. The details of the procedure have been described previously [1,2,3,4,5,6,16,17]. Briefly, 20 patients with hemophilia (HPs 1–20) were diagnosed with HIV-1 infection between 1990 and 1994. HP 21 was infected with HIV-1 via imported factor 9 before 1987 and was diagnosed in 1987. In this study, FLs were sequenced from 62 KSB-infected patients, including 3 plasma donors (O, *P*, and R) and 20 patients with hemophilia (designated as HP 1–20), 1 infected with CRF02-AG, 1 infected with subtype D, and 1 infected with subtype B (Appendix A). In this study, the FL sequence denotes the length from the start codon of Gag to Nef with the terminal stop codon.

### 2.3. RNA/DNA Preparation and FL Gene Amplification

Blood samples were collected from 20 HPs at 6-month intervals for CD4+ T cell measurements. Sera were used before the year 2000, and peripheral blood mononuclear cells (PBMC) were used after 2000 for PCR amplification.

Total RNA was extracted from 300 μL serum samples using a QIAamp UltraSens Viral RNA kit (Qiagen, Hilden, Germany), and 2 μL aliquots of RNA were reverse transcribed by mixing with 1 μL of oligo(dT), 1 μL of dNTPs, and 6 μL of DEPC treated water, followed by incubating the mixture at 65 ℃ for 5 min and then on ice for 1 min [2,3,4,5,6]. Next, 4 μL of 5× buffer, 2 μL of 0.1M DTT, 3 μL of DEPC-treated water (Ambion Inc., Foster City, CA, USA), and 1 μL of Superscript III reverse transcriptase (Invitrogen, Carlsbad, CA, USA) were added to each sample. The samples were incubated for 50 min at 50 ℃. The reaction was terminated by incubation for 5 min at 85 ℃ and then on ice for 1 min.

DNA was extracted from 400 μL of PBMC samples using a QIAamp DNA Mini kit (Qiagen, Hilden, Germany), and 5 μL aliquots of DNA were used for the nested PCR.

The *vif, vpr, tat/rev*, and *vpu* regions (1.2-kb) were amplified via nested PCR using TaKaRa r-taq (Takara Bio Inc., Shiga, Japan). The first and second PCRs were performed in 20 μL and 50 μL reaction mixtures, respectively. The outer primer pairs used were 545 and KMK2, whereas the inner primer pairs used were 548F and LA106 (Table 1). After an initial denaturation at 95 ℃ for 10 min, 35 cycles were run at 95 ℃ for 30 s, at 52 ℃ for 30 s, and at 72 ℃ for 2 mins and 30 s, followed by a final extension step at 72 ℃ for 10 mins. The second PCR was performed with 1 μL of the first PCR product; the cycling conditions were set as follows: 95 ℃ for 30 s, 57 ℃ for 30 s, and 72 ℃ for 1 min and 30 s, as well as a final extension at 72 ℃ for 10 mins. The procedure used for the amplification of the remaining genes is described elsewhere [2,4,5,6]. A maximum of five PCR, including that using a negative control, were performed per sample at a given time. To avoid selection bias, all positive PCR amplicons were sequenced and used for FL. Amplification was performed via five overlapping PCR steps (Table 1) [16,17], followed by direct sequencing using an Applied Biosystems 3730XL DNA Analyzer (Foster City, CA, USA) [5].

### 2.4. Phylogenetic Tree Analysis

In total, 70 FL sequences were obtained from 20 patients with hemophilia. The sequences from 42 local controls and 1 subtype B infected patient were aligned against the HIV-1 subtype reference dataset from the HIV Sequence Database (http://hiv-web.lanl.gov/content/hiv-db/Subtype_REF/align.html) (accessed on 20 July 2020) Phylogenetic trees were constructed using IQ TREE with 1000 bootstrap replicates [18].

### 2.5. Viral Signature Pattern Analysis (VESPA)

The VESPA program (http://www.hiv.lanl.gov/content/sequence/VESPA/vespa.html) (accessed on 20 July 2020). was used to identify the sites within each sequence group that was distinct from those in other groups [19].

### 2.6. Statistical Analysis

Data are presented as means ± standard deviation. Statistical significance was determined using Student’s two-tailed *t*-tests, paired *t*-test, chi-square tests, Fisher’s exact test, and Pearson’s correlation coefficient using MedCalc. Results were deemed statistically significant when the *p*-value was <0.05.

### 2.7. Nucleotide Sequence Data

The GenBank accession numbers for the sequences in this study are listed as follows: AF224507, AY839827, DQ054367, DQ295192-96, DQ837381, JQ316126-37, JQ341411, JQ429433, KF561435-43, KJ140245-66, KU869610, MK577478-81, MK871374, MG461319-22, MN237642-46, MN043576-607, MT101871, MT224125-27, MT559044-66, MT582420-24, MT679550-53, MW405263-343, and MW881608-49. Subtype B sequences were randomly retrieved from the Los Alamos National Laboratory (LANL) HIV Database.

## 3. Results

### 3.1. Origin of the KSB of Subtype B

A major strength of this study is that the sequences of the earliest KSB-infected patients were included. We found that patient BGO, who was diagnosed in July 1988, was the first patient infected with KSB (MT559045). Hence, we obtained the FLs in 36, 23, and 3 patients diagnosed in 1988–1991, 1992–1993, and after 1993, respectively (Appendix A and Figure 1). These 36 patients who were diagnosed in 1988–1991 correspond to 84% of all 43 KSB-infected patients diagnosed in this period [4].

### 3.2. Molecular Epidemiologic Data on the FL HIV-1 Gene

In 20 HPs, 71 FLs were obtained at 60 time points over 287 ± 99 months (about 24 years) from the outbreak, in January 1990. We obtained 169 FLs from 65 patients. Of these, the sequences were obtained from 2 or more samples collected on different dates from 30 patients, including 19 patients with hemophilia (Appendix A). Phylogenetic analysis revealed that the earliest 62 FLs obtained from 62 patients (20 HPs and 42 local controls) belonged to KSB, whereas 2 FLs obtained from 2 patients belonged to subtypes B and D (Figure 1). The 62 KSB FLs were subdivided into several clusters, including two large clusters (“O,” which comprised 9 sequences and “P,” which comprised 13 sequences) that included 20 HPs and plasma donors O and *P*. The bootstrap values of the nodes for clusters O, *P*, and KSB were all 100%, as determined using 1000 bootstrap replicates (Figure 1).

### 3.3. Korean Signature Pattern Amino Acid Residues

We previously reported the signature pattern of amino acids at residues 12 and 26 in the Gag and Env proteins [5,6]. Additionally, 8 and 9 signature pattern amino acids were determined in the Vif and Nef proteins, respectively [20,21]. We found 31 novel Korean signature nucleotides in the *pol* gene based on our previous report [2] compared with 31 sequences from 15 subtype B infected Korean patients (Appendix A). Of those, 11 were nonsynonymous substitutions, and 20 were synonymous substitutions, compared to those in subtype B.

Overall, in the FL sequences that were over 8609–8618 bp long, the signature pattern analysis indicated 91 signature nucleotides (16, 21, 6, 2, 2, 2, 35, and 7 in *gag*, *pol*, *vif*, *vpr*, *tat/rev*, *vpu*, *env*, and *nef* genes, respectively; 1.1%) that distinguished 20 HPs and 42 local controls within KSB (*p* < 0.05). In total, 48 and 43 signature nucleotides originated from clusters O and *P*, respectively. Of those, only six positions in *gag*, *pol, vif,* and *env* genes contained 100% specific nucleotide(s) positions in clusters O and *P* [2,3,4,5,6], compared to 0% in local controls (Appendix A).

### 3.4. Sequence Identities of HPs Compared to Plasma Donors O and P

The earliest FLs from the donors O and *P* were 8609 bp and 8618 bp long, respectively. The sequence identity between the earliest sequences determined in October 1991 and the last sequences determined in January 2002 (8627 bp) from donor O was 96.5%. The sequence similarity in cluster O between the earliest sequences from donor O and the sequence of each HP averaged 97.7% ± 0.8%. In cluster *P*, the sequence similarity between the earliest sequences from donor *P* and each HP was 97.4% ± 1.4%.

We determined the correlation between the sampling intervals after the outbreak (January 1990) and the number of nucleotide differences observed, relative to those of the corresponding plasma donor. Four patients who were first sampled in 2002 exhibited the lowest sequence identity (Figure 2A). The sequence identity significantly dropped over time since the outbreak. The lowest sequence identity was 89.8% at 153 months in October 2002 in HP-20. The overall correlation coefficient, γ, was estimated to be −0.85 in 19 HPs (*p* < 0.001) (Figure 2A).

When we analyzed 40 FLs before the administration of cART, the γ value was 0.77 (*p* < 0.001) (Figure 2B), whereas at the time of administration of cART, the γ value was−0.17, compared with that observed just before the administration of cART (Figure 2C). In the same context, the correlation between the sampling year and sequence length before the administration of cART was also significant in 20 HPs (*r* = 0.79; *p* < 0.001) (Figure 2D).

In addition, we found this phenomenon in 42 local control patients without hemophilia (*r* = 0.38; *p* < 0.01) (Figure 2E), and the correlation was higher in 12 local controls with sequences determined at 2 time points before cART (*r* = 0.61; *p* = 0.001).

### 3.5. Sequence Length of HIV-1 KSB Significantly Increases over Time before cART Administration

Among the earliest 20 FLs in 20 HPs, we excluded 4 HPs because the sequences were obtained after 10 years since the outbreak in January 1990. The earliest sequences obtained over 32 ± 5 months from the outbreak revealed 8614 ± 17 nucleotides (*n* = 16) with an increase of 4.1 ± 16 nucleotides per year compared to the corresponding sequences of the donors. The second sequences obtained more than 5 years after the outbreak were determined from samples collected from 17 HPs before cART (*n* = 23). This revealed 8650 ± 22 nucleotides with a significant increase of 39 ± 22 nucleotides over 143 ± 41 months from the outbreak (*p* < 0.0001); however, the first sequences of donors O and *P* obtained in 1991 and 1993 were used.

We found that the sequence length increased significantly over time before cART (Figure 2B). The sequence length by gene was analyzed to confirm which gene demonstrated an increase in sequence length. There was a significant increase in the sequence length of *env* (*p* < 0.001) and *nef* (*p* < 0.05) genes (Table 2). This increase originated in the variable regions of both genes (data not shown).

In contrast, we analyzed the changes in sequence length within each HP. The sequence length increased by 39 ± 30 nucleotides over 110 ± 43 months between the first and last sequences in 16 HPs, with sequences determined at ≥2 time points before cART (*p* = 0.0001) (Table 2). This corresponds to an annual increase of 4.4 nucleotides and a requirement of 47 and 32 years to reach the mean length of SIVsm and HIV-2, respectively.

The sequence length in the *env* gene increased by 31.7 ± 25 nucleotides during the same period (*p* = 0.0001).

In contrast, the sequence length decreased by 8.0 ± 23 nucleotides over 181 ± 47 months at the administration of cART in 16 HPs, compared to the sequence length determined just before the administration of cART (Appendix A).

To confirm whether this phenomenon occurs in patients without hemophilia, we analyzed the sequences in 12 KSB-infected patients at ≥2 time points among 42 local controls. There was also a significant increase in sequence length between the FLs over 104 ± 43 months (33 ± 29 nucleotides) (*p* < 0.01). The increase in the *env* sequence length (24 ± 25 nucleotides) over 104 ± 43 months (*p* < 0.05) significantly contributed to the 33 ± 29 nucleotide increase, whereas the increase in sequence length of the *nef* gene was not significant.

Combining 20 HPs with 42 local controls, we observed significant correlations between 100 FLs and the sampling year (*p* < 0.001) (Figure 2F), and between the sequence length of *env* and the sampling year (*r* = 0.45, *p* < 0.001); no significant correlation was observed with cART.

Additionally, we found a significant correlation even in individual patients with long-term slow progression (Figure 3).

### 3.6. The Sequence Length in HIV-1 Subtype B Significantly Increases over Time

To confirm whether the sequence length increases in subtypes other than KSB, we randomly selected 64 FLs (subtype B) from the LANL HIV Database from 1983 to 1998 before cART. The correlation between the sampling year and the sequence length was significant for FLs (*r* = 0.43, *p* < 0.001), *env* (*r* = 0.26, *p* < 0.05), and *nef* (*r* = 0.30, *p* < 0.05) genes (Figure 4A–C).

### 3.7. Clinical Significance of the Increase in Sequence Length in HIV-1 Subtype B

Several studies have reported that the V1-V2 envelope loops and the *env* sequence length increase by approximately 1% per year in the early phases of typical infections [24,25,26]. There had been several reports on the elongation of the V2 region in long-term nonprogressors (LTNPs) [27,28,29,30]. We analyzed the correlation between the number of amino acids in the V2 region and the duration since the diagnosis of KSB and the subtype B infection. Consistent with the increase in sequence length over time, there were significant correlations between the number of amino acids in the V2 region and the duration of the infection (*n* = 213, *r* = 0.60, *p* < 0.001) as well as between CD4+ T cell count and the number of amino acids in the V2 region in 75 patients (*n* = 213, *r* = −0.17, *p* < 0.05). Additionally, there were significant inverse correlations between the FLs and the CD4+ T cell counts > 100/μL in all 65 patients (*r* = −0.30, *p* = 0.001) and between the sequence length of the *env* gene and the CD4+ T cell counts > 100/μL before cART (*r* = −0.34, *p* < 0.001) (Appendix A).

## 4. Discussion

This study provides evidence suggesting that the sequence length increases over time, as demonstrated by extensive sequence data collected from 20 patients with a well-known history of hemophilia and KSB-infected local controls. Owing to well-documented primary HIV-1 infection [1,2,3,4,5,6,7] and extensive sequence data obtained for over 30 years from 20 HPs with a common source of HIV-1, we were able to confirm that the FL significantly increases over time; the strength of the correlation was stronger in 20 HPs (Figure 2D) than that in local controls (Figure 2E). Otherwise stated, the more homogeneous the cohort, the higher ts the strength of the correlation.

To our knowledge, this is the first report on the association between the sequence length in FLs and the duration of infection. At any given time, viral populations will be dominated by those strains that are the best fit at that time [27]. However, several reports have focused on V1 and V2 elongation in an elite controller [28,29,30,31] and on the consistent usage of the CCR5 coreceptor [32]. The sequence length of the *nef* gene in HIV-1, a virulence gene, is significantly shortened compared to that of other genes SIVsm [33] (Appendix A), implying pathogenicity [34].

It is important to reemphasize that viruses adopt a symbiotic pathway rather than killing the host or evolving toward attenuated pathogenicity [27]. In fact, the replicative capacity of HIV-1 in the 2000s was significantly lower than that of the virus in the 1980s [35]. Consequently, the survival period was longer in people infected in the 2000s; however, various factors might be involved. The order of replicative fitness by subtype is as follows: D > A > C [36]. Based on this, the increase in sequence length could be considered as a strategy or evolutionary direction for the virus to adapt under the pressure exerted by the immune system of the host and to coexist with the host.

The increase in sequence length implies that the longer the coding sequence (CDS) length, the lower is the density of ribosomes, resulting in less efficient protein synthesis [37] and fewer virus copies. This may be why an increase in sequence length, such as V1 or V2 elongations, occurs in the elite controller or LTNP [27,28,29], as observed in this study. In the same context, the prognosis of patients with SIVsm and HIV-2 exhibiting longer CDSs is better than that of patients with HIV-1. The increase in CDSs in HIV-1 over time is likely to be related to an attenuated pathogenicity and a different evolutionary direction.

This study had a few limitations. First, sera sampled from different individuals at various time points were used, and the time lag between the primary infection and sampling was particularly long (over 10 years in four HPs). Second, owing to the lack of sample availability, before 2000 and after 2000, sequences were obtained from sera and PBMCs, respectively. However, to our knowledge, the sequence length of HIV-1 is not affected by the sample type, although the sequence identity is affected by G-to-A hypermutations due to ApoBec3 proteins [38]. In addition, we confirmed that there was no difference in sequence length of the *nef* gene in the samples used for RT-PCR and nested-PCR, which were obtained on the same day. Therefore, we do not think that the two distinct methodologies have impacted the data or the conclusions.

In this study, the sequence length of HIV-1 increased by 4.4 nucleotides per year before cART. When viruses were transmitted to another patient, among quasispecies, the possibility of infection by HIV-1 with a shorter sequence might be higher than that by HIV-1 with long sequences. Thus, at the population level, the accumulation effect of the increase in sequence length might be slower than that at the individual level because of the bottleneck effect. Our novel data suggest that the increase in CDSs over time might indicate an evolutionary direction.

## Figures and Tables

**Figure 1 viruses-13-00806-f001:**
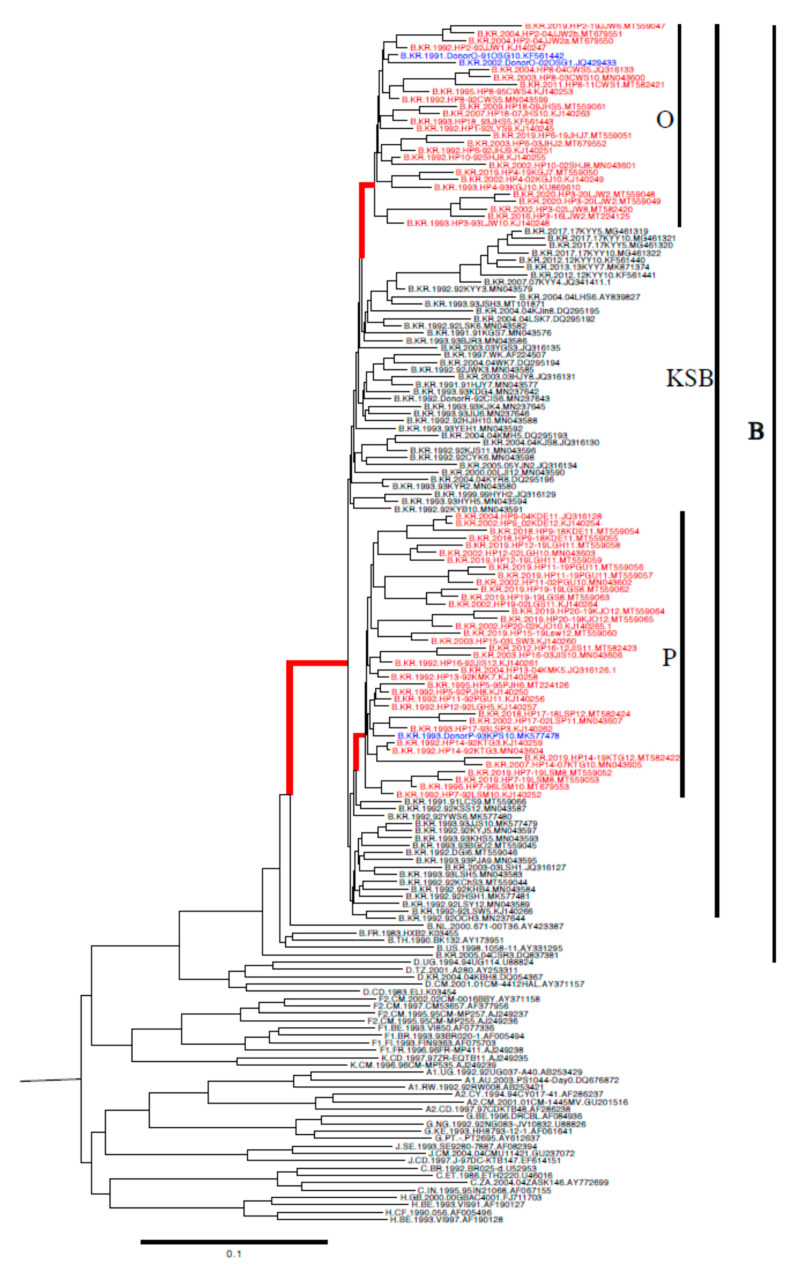
Phylogenetic tree analysis of the earliest near full-length sequences (about 8615 bp from the *gag* to the *nef* genes) of 64 Korean patients infected with HIV-1 performed using IQ TREE with 1000 bootstrap replicates: 20 patients with hemophilia (HPs); 3 plasma donors (O, *P*, and R); 39 local controls infected with the Korean subclade of HIV-1 subtype B (KSB) and 2 non-KSB-infected patients. The upper 103 sequences belonged to KSB, and 2 sequences (05CSR3 and 04KBH8) belonged to subtypes B and D, respectively. In total, 9 and 13 patients, including donors O (Cluster O: donor O, 1–4, 6, 8, 10, and 18, as designated by the taxa in red) and *P* (Cluster *P*, as designated by the taxa in red), were strongly clustered within the KSB-infected local controls. The two digits before the ID of each patient and the one or two digits after the ID of each patient denote the year and month of sampling, respectively. The bootstrap values of the nodes for clusters O, *P*, and KSB (designated by bold in red) were 100%, as determined by 1000 bootstrap replicates.

**Figure 2 viruses-13-00806-f002:**
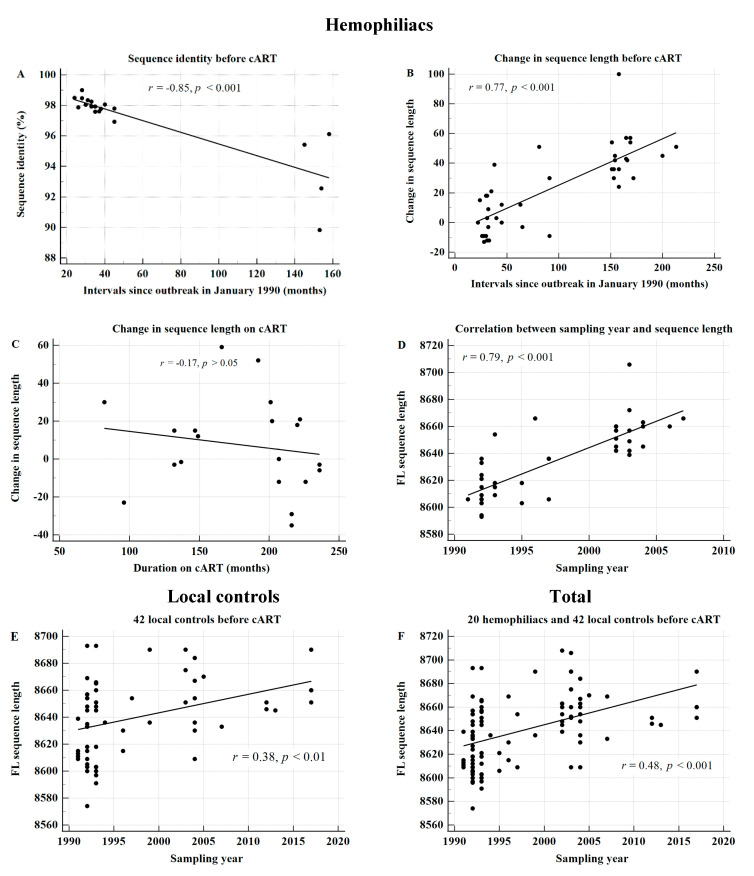
The full-length (FL) sequence length increases over time and the sequence identity decreases over time in KSB-infected patients. (**A**–**C**): Sequence identity and FL sequences according to the interval since the outbreak in January 1990 in 20 HPs. (**A**) Correlation between the intervals from the outbreak to sampling and the sequence identity of each earliest FL sequence before cART in the 20 patients with hemophilia, compared with the corresponding earliest sequence of the plasma donors. (**B**) The correlation coefficient (CC), γ, was 0.77 for the 39 FLs before cART (*P* < 0.001). (**C**) The CC, γ, was −0.17 upon cART. (**D**–**F**), Significant increase in the FL sequence length by sampling year before cART, in patients with hemophilia (**D**), 42 KSB-infected local controls (*n* = 60) (**E**), and all the 62 KSB-infected patients (**F**).

**Figure 3 viruses-13-00806-f003:**
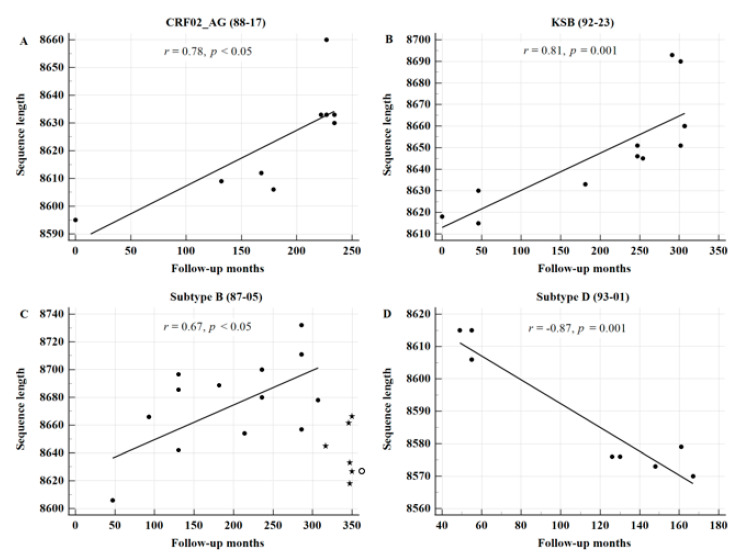
The length of FLs increased over time in 3 long-term nonprogressors (LTNPs), whereas it decreased in subtype D-infected patients. The correlation coefficient between the sequence length and the duration of infection was significant in 3 LTNPs over 25 years (up to 294, 307, and 286 months). Patients infected with CRF02_AG (**A**), KSB (**B**), and subtype B (**C**) were diagnosed with HIV-1 infection in 1988, 1992, and 1987, respectively. Interestingly, the sequence length in patients infected with subtype B significantly decreased from 8697 ± 26 (*n* = 6), just before 286 months (March 2011), to 8623 ± 24 (*n* = 6; marked as black stars), after 286 months, when the plasma RNA copy number significantly increased (*p* < 0.001) [22]. The sequence length significantly decreased after 307 months. In contrast, it was inversely significant in patient 93-01, who was diagnosed in December 1992 (**D**). The first sexual contact and diagnosis of pulmonary tuberculosis of the patient were observed in 1988 and 1989, respectively. He was treated with Korean Red Ginseng (12,720 g) from April 1993 to August 2004 [23]. Thus, despite the most rapidly progressing subtype D infection, he remained healthy for at least 12 years [23]. The correlation analysis did not include the data marked as stars and as a white circle on cART.

**Figure 4 viruses-13-00806-f004:**
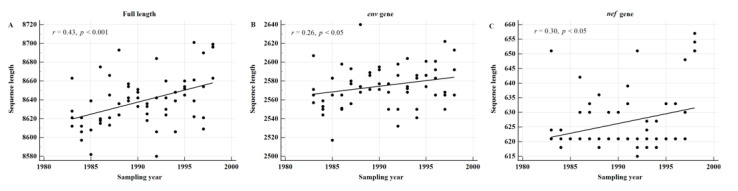
The sequence length increases over time in subtype-B-infected patients. A total of 64 FLs (subtype B) were randomly selected from Los Alamos National Laboratory Database over 16 years from 1983–1998 (evenly 4 per year). They were obtained before cART. The correlation between the sampling year and the sequence length was significant for FLs (**A**), *env* (**B**), and *nef* (**C**) genes. Of the 64 sequences, 44 originated from the United States, and if these alone are analyzed, the correlation is even higher (in order, *r* = 0.58, *p* < 0.001; *r* = 0.40, *p* < 0.01; and *r* = 0.46, *p* < 0.01 for FLs, *env*, and *nef* genes, respectively).

**Table 1 viruses-13-00806-t001:** Primer sequences used for the polymerase chain reaction.

Nested PCR	Primer	Sequences (5′-3′)	
For the full-length *gag* gene [6]			
First PCR	503k	5′-CCKTCTGTTGTGTGACTCTGGTAA-3′	forward
	524	5′-CATTGTTTAACTTTTGGGCCATCC -3′	reverse
Second PCR	504F	5′-TCTCTAGCAGTGGCGCCCGAAC-3′	forward
	505	5′-GAGACATGGGTGCGAGAGCGT-3′	forward
	522	5′-ACTGTCCTACTTTGATAAAACCTC-3′	reverse
For the full-length *pol* gene [2,7]			
First PCR	HXB2	5′-GTGGGAGAAATCTATAAAAGATGG-3′	forward
	OBP2	5′-GAGACTCCCTGACCCAGATG-3′	reverse
	OBP2k	5′-GAGACTCCCTGACCCAGATG-3′	reverse
	550	5′-CCTAGTGGGATGTGTACTTCTGAA-3′	reverse
Second PCR	PO1	5′-AAAATTGCAGGGCCCCTAGGA-3′	forward
	PR3-1	5′-GAAGCAGGAGCCGATAGACA-3′	forward
	OBP4	5′-CAATCATCACCTGCCATCTG-3′	reverse
	P2	5′-AGGAAGGACACCAAATGAAAG-3′	forward
	P16	5′-GGATKAGTGCTTTCATAGTGA-3′	reverse
For *vif, vpr, tat, rev,* and *vpu* genes			
First PCR	545	5′-GCAGTACAAATGGCAGTATTCATC-3′	forward
	KMK2	5′-ATGGGAATTGGTTCAAAGGA-3′	reverse
Second PCR	548F	5′-AGTGACATAAAAGTAGTRCCAAGAA-3′	forward
	LA106	5′-TTCACTCTCATTGCCACT-3′	reverse
For the full-length *env* gene [5]			
First PCR	OWE1	5′-TCATCAAGTTTCTCTATCAAAGCA-3′	forward
	OWE2	5′-TCTGACTGGAAAGCCCACTT-3′	reverse
Second PCR	OWE3	5′-GCAATATTAGCAATAGTTGTGTGG-3′	forward
	OWE4	5′-ATACTGCTCCCACCCCTTCT-3′	reverse
For *nef* gene [4]			
First PCR	Nef5′5	5′-AGGATTGTGGAACTTCTGGGAC-3′	forward
	LTR3	5′-AGGCTCAGATCTGGTCTAAC-3′	reverse
Second PCR	Nef3	5′-ATGGGTGGCAAGTGGTCAAA-3′	forward
	N10	5′-CGTCCAGAATTCGGAAAGTCCCCAGCGGAAAGT-3′	reverse

**Table 2 viruses-13-00806-t002:** Changes of the sequence length of HIV-1 over 110 ± 43 months by genes in patients with hemophilia.

Sequences with Intervals	*Gag*	*Pol*	*Vif*	*Vpr*	*Tat*	*Rev*	*Vpu*	*Env*	*Nef*	FL
First sequences (*n* = 16)	1503 ± 0	3012 ± 0	579	291	215	76	247 ± 2	2563 ± 19	622.5 ± 2	8620 ± 20
Second sequences (*n* = 16)	1504 ± 3	3011 ± 3	579	291	215	76	247 ± 2	2595 ± 22	627.2 ± 8	8659 ± 23
*p*-value	NS	NS	NS	NS	NS	NS	NS	< 0.001	< 0.05	= 0.0001

There was an interval of approximately 110 ± 43 months between the sequences obtained at the two time points. NS = not significant. *p*-value calculated using a paired *t*-test. Data are expressed by mean ± standard deviation.

## Data Availability

All data are available from the corresponding author on request.

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
