# Peer review of "Sequence Length of HIV-1 Subtype B Increases over Time: Analysis of a Cohort of Patients with Hemophilia over 30 Years"

_viruses, 2021, doi:10.3390/v13050806_

Round 1

Reviewer 1 Report

The Authors addressed adequately the critiques.

Author Response

There was no comment from reviewer 1.

Reviewer 2 Report

I agree with the adaptations made but still think that the conclusions contains too little reflections on the possible biases and shortcomings of the study.

Author Response

  1. A major concern is that plasma was used to sequence virus before the the year 2000 while PBMCs were used from the year 2000 onwards. My understanding is that the method used to purify DNA from PBMCs will yield both unintegrated HIV and chromosomal DNA (containing proviral HIV DNA), the latter may be archival and have an impact on the interpretation of the data. This is a major short coming of this study, which while mentioned in the Discussion, lacks sufficient reflection on how using two distinct methodologies might impact on the data and the conclusions of the study.

We have presented the following information as a limitation in the Discussion (lines 323-328):

“Second, owing to the lack of sample availability, before 2000 and after 2000, sequences were obtained from sera and PBMCs, respectively. However, to our knowledge, the sequence length of HIV-1 is not affected by the sample type, although the sequence identity is affected by G-to-A hypermutations due to APOBEC3 proteins [38]. In addition, we confirmed that there was no difference in sequence length of the nef gene in the samples used for RT-PCR and nested-PCR, which were obtained on the same day. Therefore, we do not think that the two distinct methodologies have impacted the data or the conclusions.

  1. For Figures 2E and F, please clarify in the figure legend if the data analyzed is pre -cART? Also please specify if they are FL sequences and/or env.

We have added “FL” to the Y-axis, in the legend (lines 202-08), and “before cART” in the title of Figure 2E and F.

  1. Line 265. Please clarify in the manuscript if the sequences obtained from the Los Alamos database are from cART naive individuals?

cART was introduced for treatment in 1997. We confirmed, using a PubMed search, that the 8 GenBank accession numbers used in this study (originated in 1997 and 1998) were of individuals not exposed to cART.

We have added “before cART” in line 264 and “They were obtained prior to cART.” in lines 271 in the legend.

  1. Signature patterns of amino acid residues are noted for various HIV genes including pol. Please detail in the discussion how these signatures differ from those reported in a previous publication on what appears to be the same cohort (Cho et al 2011 AIDS Res Hum Retroviruses 27:613).

For clarity, we added “based on our previous report [2]” in line 164 as follows; “We found 31 novel Korean signature nucleotides in the pol gene based on our previous report [2] compared with 31 sequences from 15 subtype B-infected Korean patients”.

We did not compare the Korean signature amino acids in KSB with those of the Western subtype B in the pol gene in 94 patients, including 20 patients with hemophilia in a previous study [2]. We only compared them between the clusters O/P and KSB-infected local controls. To date, the Korean signature amino acids in the pol gene have not been reported (except for references 5, 6, 20, and 21). Thus, we mentioned this based on a previous study [2] in Table S2 because the number of patients in the previous study [2] was higher than that of KSB-infected patients in the present study (62 patients).

Therefore, we do not believe that mentioning the Korean signature amino acids in the Discussion is necessary.

  1. Lines 221-222. P values for env and nef in the text do not correspond to values in Table 2? It is not clear if the values are SD or SEM in the table. The data, despite the p values, do not appear convincing if SEMs?

We have made the necessary corrections and added the following section in the footnote (line 226): “Data are expressed as mean ± standard deviation.”

  1. Lines 266-267. the data (graphs) for env and nef should appear in the results or in the supplementary section.

We have added the graphs for env and nef genes in Figure 4. We also added the following section to the legend (lines 271-274):

They were obtained prior to cART. The correlation between the sampling year and the sequence length was significant for FLs, env, and nef genes. Of the 64 sequences, 44 originated from the Unites States and, if these alone are analyzed, the correlation is even higher (in order, r = 0.58, p < 0.001; r = 0.40, p < 0.01; and r = 0.46, p < 0.01 for FLs, env, and nef genes, respectively).”

  1. line 279 - please specify the factors in this sentence being referred to rather than stating aforementioned factors.

We have clarified that the factors are “the number of amino acids in the V2 region and the duration of infection” (lines 282-283).

  1. The discussion contains data not present/mentioned in the results section (i.e Figure S2 and relevant text) and specifically data on other clades which was recommended previously to be deleted. This should be removed from the Discussion as well as the corresponding Figure in the supplementary section

We have deleted Figure S2 and added the corresponding text (lines 301-315 in the previous version) to the Discussion.

Lines 301-315;

The decreasing rate of CD4+ T cell counts is faster in subtype D than that in subtype B [15, 33]. FL in subtype D was significantly shorter than those in subtypes B, KSB, C, and G (Figure S2). This might be associated with the shorter survival time required for subtype D viruses to adapt in vivo than that required for other subtypes. Thus, there was no correlation between the sampling year and sequence length in subtype D (data not shown). When we compared the sequence length in the env gene among three subtypes, sequence length in subtype B (n = 64) was significantly longer (2,575 ± 21) than 2,547 ± 20 in subtype D (patient 93-01) (P < 0.0001). FLs in subtypes B (8,639 ± 28) were significantly longer than those in subtype D (n = 10, 8,589 ± 19, P < 0.0001).

It is known that sequence length is significantly longer in SIVsm than in HIV-2. The increase results from the increase in env and nef genes. Taken together, these data suggest that the longer sequence length (SIVsm > HIV-2 > SIVcpz > HIV-1 subtypes B/KSB > D) (Figure S2) corresponds to a longer survival. When we translated the longevity of chimpanzees and sooty mangabeys into human beings (60 years), there was a significant correlation between the sequence length and survival duration (r = 0.90 ~ r = 0.88, P < 0.05).

Round 2

Reviewer 2 Report

No further comments

This manuscript is a resubmission of an earlier submission. The following is a list of the peer review reports and author responses from that submission.

Round 1

Reviewer 1 Report

In the current manuscript, Cho et al describe sequence of HIV increases over time and proposes such an increase leads attenuation.  Experiments for sample analysis collected over several years are straight forward and are mainly based on genome sequence. The data collected can be informative and could be a base for further analysis whether such an increase full length coding sequence have important implication in the level HIV pathogenicity.  Although informative the implication that such phenomenon leads to attenuation may pre-matured and requires additional studies. 

Other points

How do we know each of the genome sequence analyzed are infectious or not to lead to the conclusions described?

Could this be coincident artifacts during reverse transcription and replication of genome overtime that have no implication in HIV pathogenicity or attenuation?

As the author pointed out sequence of the LTR is missing.  Could that have importance for the current manuscript?

What would be the explanation that the increase seen in one subtype but not in the other?

The section Sequence length of HIV-1 is significantly shorter than in HIV-2 and SIVsm is not new information and is existing knowledge.  The sequence differences in these strains are significant and distinct.  Such comparison and conclusion with the other section of the manuscript could be misleading. 

Reviewer 2 Report

In this manuscript Cho and co-authors performed full HIV-1 genome sequencing of a selection of Korean subtype B infected patients including a number of patients in two transmission clusters resulting from the distribution of contaminated blood to patients with hemophilia. They supplemented their dataset with a selection of full genome sequences from the Los Alamos database. Their main conclusions were that the overall sequence length of HIV-1 in increasing over time and that longer sequence length is associated with a slower disease progression.

The authors have performed a large amount of work and the conclusion of their investigation, if correct, is important. However, it is my opinion that the data they provide are insufficient to support the firm statement about the genome length. I also found it a very difficult article to read.

Major remarks

It is very difficult throughout the text to discriminate between data and conclusion drawn from the Korean samples and data and conclusions based on the Los Alamos sequences. This is to my opinion an important aspect. I also don’t see the point of including the analysis of different subtypes. The conclusions drawn from the subtype specific analysis will only be valid if a sampling bias can be excluded. No information is provided on how the Los Alamos sequences are selected and on how selection bias was avoided or minimalized. It would have been better in my opinion if the authors only concentrated on subtype B.  

The material and method section overall is too concise, description of the methods used is too vaque. Information of the sample and sequence selection too limited.

If I understand it correctly, for the Korean patients, only a single sequence at a single time point was generated. In that case it the term ‘longitudinal analysis’ is a bit confusing.

In the material and method paragraph 2.2 it is written that sera were collected before 2000 and BMC after 2000. Does that mean that the target material for sequencing differs over time? If this is the case it may seriously impact the results as it is known that the vast majority of proviral sequences are deficient so comparison of viral sequences from the plasma and blood compartment is at least tricky. The title of paragraph 2.2 is ‘RNA preparation’ and only the procedure for RNA preparation, reverse transcription and sequencing is described. What about the PBMC? Where the samples collected on-ART majority PBMC samples? If not, then the possibility of amplifying virale RNA from the plasma of these patients suggests treatment failure. The viral population present then may be a population under higher selection pressure. This may influence the results and conclusions.